# Effect of Green Light Replacing Some Red and Blue Light on *Cucumis melo* under Drought Stress

**DOI:** 10.3390/ijms25147561

**Published:** 2024-07-10

**Authors:** Xue Li, Shiwen Zhao, Qianqian Cao, Chun Qiu, Yuanyuan Yang, Guanzhi Zhang, Yongjun Wu, Zhenchao Yang

**Affiliations:** 1College of Horticulture, Northwest A & F University, Xianyang 712100, China; 2Key Laboratory of Northwest Facility Horticulture Engineering of Ministry of Agriculture and Rural Affairs, Xianyang 712100, China; 3College of Life Sciences, Northwest A & F University, Xianyang 712100, China

**Keywords:** melo, drought resistance, light quality, LED

## Abstract

Light quality not only directly affects the photosynthesis of green plants but also plays an important role in regulating the development and movement of leaf stomata, which is one of the key links for plants to be able to carry out normal growth and photosynthesis. By sensing changes in the light environment, plants actively regulate the expansion pressure of defense cells to change stomatal morphology and regulate the rate of CO_2_ and water vapor exchange inside and outside the leaf. In this study, *Cucumis melo* was used as a test material to investigate the mitigation effect of different red, blue, and green light treatments on short-term drought and to analyze its drought-resistant mechanism through transcriptome and metabolome analysis, so as to provide theoretical references for the regulation of stomata in the light environment to improve the water use efficiency. The results of the experiment showed that after 9 days of drought treatment, increasing the percentage of green light in the light quality significantly increased the plant height and fresh weight of the treatment compared to the control (no green light added). The addition of green light resulted in a decrease in leaf stomatal conductance and a decrease in reactive oxygen species (ROS) content, malondialdehyde MDA content, and electrolyte osmolality in the leaves of melon seedlings. It indicated that the addition of green light promoted drought tolerance in melon seedlings. Transcriptome and metabolome measurements of the control group (CK) and the addition of green light treatment (T3) showed that the addition of green light treatment not only effectively regulated the synthesis of abscisic acid (ABA) but also significantly regulated the hormonal pathway in the hormones such as jasmonic acid (JA) and salicylic acid (SA). This study provides a new idea to improve plant drought resistance through light quality regulation.

## 1. Introduction

Drought is currently one of the most serious global problems threatening crop growth and productivity [1], and its global crop losses may exceed those caused by all other abiotic stresses combined [2,3]. In response to environmental stimuli, plants balance water loss by optimizing stomatal opening and closing [4].

Stomata consist of paired guard cells in the plant epidermis that act as windows between the plant and the external environmental gases, and plants respond to various environmental stimuli through changes in guard cell volume expansion, which leads to stomatal opening and closing [5]. Stomatal opening increases photosynthetic CO_2_ uptake to promote plant growth and dissipates water through transpiration [6]. Stomatal movement is regulated by various factors such as light, temperature, and CO_2_ [7,8,9]. Therefore, it has been proposed to improve photosynthesis and water use efficiency by changing environmental conditions so that the stomatal response rate is increased [9,10,11,12]. However, by increasing CO_2_ conductance by increasing stomatal conductance, this will inevitably cause a synergistic increase in transpiration rate, which will result in a large amount of water dissipation from the plant, and when the leaf water potential is reduced to a certain threshold, the leaf will close the stomata to prevent further water dissipation [13,14,15]. Therefore, the synergy between stomatal conductance and leaf water transport capacity is important to maintain the balance of leaf water supply and loss and to ensure that leaf tissues undergo normal photosynthesis [16,17,18,19].

Cryptochrome and photosensitizer work in concert to mediate stomatal opening [20,21]. Under low doses of blue light, cryptophycin and photosensitizer act on COP1 and abscisic acid (ABA), respectively, to activate guard cell plasma membrane H^+^-ATPases. These H^+^-ATPases pump H^+^ to the outside of the plasma membrane, resulting in a positive (external) to negative (internal) potential gradient that leads to the influx of K^+^ ions into the membrane. Green light, on the other hand, inactivates cryptochrome [22], thus eliminating the inhibition of ABA production signals in guard cells and promoting a reduction in stomatal aperture. Furthermore, Frechilla et al. [23] found that green light at 490 to 580 nm reversed blue-light-induced stomatal opening in isolated epidermis, which is consistent with the results of a study by Talbott et al.’s findings [24]. In both studies, the degree of reversal was irradiance-dependent at continuous irradiance, with complete reversal when green light was twice as much as blue light. Further studies have shown that the same response can be observed in Arabidopsis leaves [25], and the work of [26] also concluded that green light (λ500–600 nm) exposure reversibly reduces stomatal conductance in lettuce. Green light signals play an important role in controlling stomatal aperture and the balance between water loss and CO_2_ uptake within the canopy, so the antagonistic nature of the green light response to blue light implies that the stomata are responsive to the B:G ratio, which should provide signals that can be fine-tuned in a way that may be beneficial to the plant in balancing the efficiency of photosynthesis and water use within the dense canopy. The present study investigated the effects of different red/blue/green ratios on photosynthesis and stomatal characteristics of melon (*Cucumis melo*) seedlings, aiming to provide some implications for improving plant water utilization by altering light quality.

## 2. Results

### 2.1. The Effect of Green Light Replacing Some Red and Blue Light on the Morphology of Melon Seedlings under Drought Stress

Seedling morphological indexes were determined on the 9th day of drought, and as shown in Table 1, plant height, fresh weight, dry weight, and total leaf area changed significantly with the increase in green light percentage. Plant height was significantly higher in the T3 treatment than in the rest of the treatments. The plant height at T3 was significantly higher than the rest of the treatments, and the plant height at T3 was significantly increased by 28.90%, 27.09%, and 13.47% compared to the CK, T1, and T2 treatments, respectively. Fresh weight was significantly higher in the T3 treatment than in the rest of the treatments. The fresh weight under the T3 treatment was significantly higher than the rest of the treatments, being 55.50%, 61.56%, and 33.69% higher than the CK, T1, and T2 treatments, respectively. The fresh weight under the T3 treatment was significantly higher than the CK, T1, and T2 treatments by 55.50%, 61.56%, and 33.69%. The dry weight was highest under the T3 treatment, which was significantly higher than that of the CK treatment by 30.15%, but there was no significant difference among the three treatments, namely, T1, T2, and T3. There was no significant difference between the three treatments of T1, T2 and T3.

### 2.2. Effect of Green Light Replacing Some Red and Blue Light on Photosynthetic Characteristics of Melon Seedlings under Drought Stress

Transpiration rate (E), stomatal conductance (gsw), intercellular CO_2_ (Ci), and net photosynthesis rate (Pn) were measured on day 3, day 6, and day 9 of the drought treatments, respectively. E was highest and significantly higher under the CK treatment than the T2 and T3 treatments on day 3 of the treatment, but significantly lower than the remaining treatments on day 6 of the treatment (Figure 1a). At treatment d 3, gsw was significantly higher under CK treatment but significantly lower than the rest of the treatments at d6 (Figure 1b). There was no significant difference in Ci among treatments on day 3, but on day 6 and day 9, the CK treatment values were lower than the remaining treatments (Figure 1c). Pn was significantly higher under T1 and T2 treatments than CK and T3 on day 3, but on d 6 and 9, the CK treatment of Pn was significantly lower than the rest of the treatments, and there was no significant difference between T1, T2, and T3 (Figure 1d).

### 2.3. Effect of Green Light Replacing Some Red and Blue Light on Stomatal Aperture and REC of Melon Seedlings under Drought Stress

The data of stomatal measurement on the 9th day are shown in Table 2. The short-term light treatment with different wavelengths had no significant effect on the length and width of stomata under different treatments, but the effects on pore length, pore width, and pore area were quite different. The pore widths of the T2 and T3 treatments were significantly decreased by 22.66% and 33.14% compared with that of the CK treatment, but there was no significant difference between the T1 and CK treatments. The pore length of the CK treatment was significantly higher than the rest of the treatments, and the pore length of the CK treatment was significantly increased by 15.78%, 16.24%, and 12.97% compared with T1, T2, and T3 treatments, respectively; due to the differences in the pore length and width of the treatments, the pore area of the treatments also differed, in which the pore area of T2 and T3 treatments was significantly decreased by 8.31% and 8.81% compared with that of the CK treatment, respectively.

Seedling leaf electrolyte osmolality was determined at 3, 6, and 9 days (Figure 2a). The relative electrolyte osmolality of each treatment increased significantly at d 6 compared to the ground on day 3 with increasing drought treatment time. Only the T3 treatment was significantly lower than the remaining treatments at day 3. At treatments on day 6 and day 9, T1, T2, and T3 were significantly lower than the CK treatments, with the T3 treatment being the most significant.

### 2.4. Effect of Green Light Replacing Some Red and Blue Light on MDA and SOD in Leaves of Melon Seedlings under Drought Stress

The MDA content of each treatment was determined on the 9th day of drought, and T1, T2, and T3 were significantly lower than the CK treatment, and the T3 treatment was the most significant. There was no significant difference between T1 and T2 treatment groups (Figure 2b). As the proportion of green light substitution increased, the SOD activity of each treatment showed a significant increasing trend; CK, T2, and T3 were significantly lower than the T3 treatment, and there was no significant difference between T1 and T2 treatment groups (Figure 2c).

### 2.5. Effect of Green Light Replacing Some Red and Blue Light on H_2_O_2_ and O_2_^∙−^ in Leaves of Melon Seedlings under Drought Stress

With the increase in the proportion of green light substitution, the H_2_O_2_ content of each treatment showed a significant decreasing trend; T1, T2, and T3 were all significantly lower than the CK treatment, the T3 treatment was the most significant, there was no significant difference between the T1 and T2 treatment groups (Figure 3a), and the results of the DAB staining of melon seedling leaves under each treatment were consistent (Figure 3b). For the O_2_^∙−^ content of melon seedling leaves under each treatment obtained as in Figure 3c, T2 and T3 were significantly lower than the CK treatment, but there was no significant difference between the T1 and CK treatment groups, and the NBT staining results of melon seedling leaves under each treatment were consistent (Figure 3d).

### 2.6. Transcriptomic Analysis of Green Light Instead of Partial Red and Blue Light on Leaves of Melon Seedlings under Drought Stress

#### 2.6.1. RNA-Seq Analysis

The sequenced RNA-seq libraries ranged in size from 41,980,568 bp to 42,519,908 (Appendix A), and the Q30 percentage (percentage of sequences with <0.1% errors) for each library was >97.45%. Overall, the RNA-seq data were of high quality and could be used for further analysis.

#### 2.6.2. Identification of Differentially Expressed Genes (DEGs) between Groups

In order to further investigate the key genes involved in the regulation of drought tolerance in melon seedlings under different treatments, this study analyzed the transcriptome of plant leaves under CK and T3 treatments after 9 d of drought. There were a total of 1260 DEGs (Differential Expression Analysis, DEG), with 591 up-regulated DEGs and 723 down-regulated DEGs in T3 compared with the CK treatment (Appendix A), and the heatmap analysis of the differential genes is shown in Figure 4a.

#### 2.6.3. GO Pathway Annotation Analysis of Differential Genes

The top 20 GO entries with the smallest *p*-value were plotted with the GO entry in the vertical coordinate and the −log10 value of the *p* or Q value of the enrichment analysis for that GO entry in the horizontal coordinate, and the results are shown in Figure 4b. In the comparison between T3 and CK, the DNA-binding transcription factor activity (GO0003700), transcriptionally regulated DNA template (GO0006355), and alginate phosphatase (GO0004805) were the first three significantly different GO pathways. There were 134, 139, and 8 genes involved in these three significantly enriched pathways, respectively.

#### 2.6.4. KEGG Pathway Annotation Analysis of Differential Genes

To further explore the biological functions of DEGs, pathway enrichment analysis was performed based on the KEGG database, and the results are presented in the form of an enrichment scatterplot. The KEGG enrichment scatterplot showed that phytohormone signaling (cmo04075), flavonoids and flavonols biosynthesis (cmo00944), and α-linolenic acid pronuclei (cmo00592) were the three pathways with the highest enrichment degree. Among them, 21 genes were significantly up-regulated and 31 genes were significantly down-regulated in the phytohormone signaling (cmo04075) pathway; 3 genes were significantly up-regulated and 6 genes were significantly down-regulated in the flavonoids and flavonols biosynthesis (cmo00944) pathway; and 9 genes were significantly up-regulated and 6 genes were significantly down-regulated in the α-linolenosuccinate substitution (cmo00592) pathway (Figure 4c).

#### 2.6.5. Analysis of Plant Hormone Synthesis Pathways

The analysis revealed that “phytohormone signaling” was an important pathway in response to light quality changes under drought stress when green light replaced part of the red and blue light in the spectrum compared with red and blue light. The results showed that several hormone signaling pathways, including growth hormone (IAA), cytokinin (CK), gibberellin (GA), abscisic acid (ABA), salicylic acid (SA), brassinosteroid (BR), jasmonic acid (JA), and ethylene (ETH), were affected by the light quality changes. Among them, 7 out of 15 differential genes related to ABA, JA, SA, GA, and BR signaling showed an up-regulation trend under the T3 treatment compared with the CK treatment (Figure 5).

#### 2.6.6. qRT-PCR Validation

The reliability of the RNA-Seq data was verified by qRT-PCR using nine randomly selected differential genes (Appendix A). The results showed that most of the gene expression trends in the qRT-PCR data were similar to the transcriptome results, and these results confirmed the reliability of the RNA-Seq data.

### 2.7. Metabolomic Analysis of Leaf Metabolism of Melon Seedlings under Drought Stress by Replacing Part of Red and Blue Light with Green Light

#### 2.7.1. Differential Metabolite Statistics

To further investigate the changes of hormone-related metabolites in melon seedlings after the green light replaced part of the red and blue light treatments, we performed a metabolomic analysis of melon seedling leaves using LC-MS. The results of the metabolomic analysis of melon seedling leaves using LC-MS indicated that a total of 103 significantly different metabolites were detected, with 44 metabolites significantly up-regulated and 59 metabolites significantly down-regulated in the T3 treatment compared with the CK treatment (Appendix A).

The metabolites were first analyzed by PCA using the principal component analysis (PCA) model (Figure 6a), and we found that the accumulation of metabolites in the T3 and CK treatments was significantly different.

#### 2.7.2. Differential Metabolites KEGG Functional Annotation

For differential metabolites, KEGG pathway analysis was performed. The top 20 pathways with the smallest *p*-value were taken for bubble plot presentation, and the results are shown in Figure 6b. The KEGG enrichment scatter plot showed that the biosynthesis of plant secondary metabolites (ko01060), the biosynthesis of plant hormones (ko01070), and the biosynthesis of amino acid compounds (ko01230) were the three pathways with the highest degree of enrichment.

#### 2.7.3. Analysis of Differential Metabolites for Plant Hormone Synthesis

Phytohormones are widely involved in the plant response to external environmental stress, and stomatal opening and closing in plants are regulated by a variety of phytohormones. Therefore, we selected the differential metabolites within the biosynthesis of phytohormones (ko01070) for analysis, and there were 13 differential metabolites, which showed that aspartic acid and jasmonic acid were significantly up-regulated under the T3 treatment compared with the CK treatment, and citric acid, α-D-glucose, tryptophan, and AMP were significantly down-regulated under the T3 treatment compared with the CK treatment. (Figure 6c).

## 3. Discussion

### 3.1. Effects of Green Light Replacing Some Red and Blue Light on the Morphology and Photosynthesis of Melon Seedlings under Drought Stress

Light plays a significant role in plant growth, development, and stress response [27]. It is known that photoreceptors for red and blue light exist in plants and that chlorophyll in plants is effective in absorbing red and blue light. However, in recent years, studies have shown that green light also plays a key role in plant growth [28] and can improve the ability of plants to cope with abiotic stresses [29,30]. In this experiment, melon seedlings were placed under treatments with different light qualities and short-term drought stress when they grew to three leaves and one heart. The experimental results showed that green light replacing part of the red and blue light treatments could alleviate the drought stress of the plants to a certain extent and keep the melon seedlings maintaining a relatively high photosynthetic capacity. With the increase in the proportion of green light, melon seedlings had significant changes in their plant height, fresh weight, dry weight, and total leaf area, and the above indexes were significantly better than the CK treatment under the T3 treatment, which was consistent with the results of the experiments conducted by Bian et al. on tomato seedlings [29] and Ma et al. on cucumber seedlings [30].

The photosynthetic efficiency of plants is reduced under drought stress, and one of the main reasons is the decrease in gsw, which limits the gas exchange between plants and the outside world, resulting in lower uptake [31]. The experimental results of this study showed that on the 3rd day of the initial drought period, stomatal conductance decreased significantly in the treatment with the addition of green light compared to the control, which was attributed to the fact that the addition of green light reversed the blue-light-mediated stomatal opening effect [32,33], resulting in a significant decrease in gsw, which led to a decrease in E in the melon seedlings. On the 6th day of treatment, Pn, gsw, and E were significantly higher in the treatment with added green light than in the CK treatment. It was hypothesized that melon seedlings under the CK treatment were significantly affected by drought stress, and they transpirated a large amount of water at the early stage of the treatment, so there was not enough water in the plants to maintain normal photosynthesis in the late stage of the treatment, whereas the group treated with added green light had a lower rate of transpiration at the early stage, and thus there was still enough water to maintain photosynthesis in the late stage.

### 3.2. Effect of Green Light Replacing Some Red and Blue Light on Antioxidant System of Melon Seedlings under Drought Stress

A reactive oxygen species (ROS) plays a key role in plant response to abiotic stresses, and it mainly acts as a signaling molecule and plays an important regulatory role in plant adaptation to stress, but it also causes some degree of toxic effects to the plants [34]. Under drought stress, this will lead to the production of and increase in ROS in plant tissues [35]. ROS concentration will increase in cells and lead to the lipid peroxidation of membranes and may even lead to the oxidative damage of cells [36]. When drought stress leads to the destruction of biofilm structure, MDA and electrolyte permeability can be used as characterization indicators to show the degree of damage it suffers [30,37]. The results showed that on the 9th day of treatment, the green light replacing part of the red and blue light treatment showed a significant decrease in MDA and electrolyte osmolality compared with that under the control treatment, indicating that the addition of green light mitigated the degree of membrane lipid peroxidation caused by drought in melon seedlings. Compared with the CK treatment, the H_2_O_2_ and O_2_^∙−^ content of the green light treatment showed a significant decrease, and the results of DAB and NBT staining of melon seedling leaves also confirmed this result, indicating that with the increase in the proportion of green light substitution, the ROS content of melon seedlings showed a decreasing trend.

### 3.3. Effect of Green Light Replacing Some Red and Blue Light on Stomatal Morphology of Melon Seedlings under Drought Stress

Plants respond to various environmental stimuli through changes in defense cell volume expansion, which leads to stomatal opening and closing [5]. Light is one of the most dynamic environmental signals and plays a key role in photosynthesis and stomatal movement [6], so light environmental conditions can be altered thereby changing stomatal opening to improve photosynthesis and water use efficiency [9,10,11]. Red and blue light regulate stomatal movement through different regulatory pathways [9,38], and green light can reverse blue-light-induced stomatal opening [24]. Therefore, green light can be used as a signal to control stomatal transpiration and plant water consumption [39]. The experimental results showed that the addition of green light in the treatment had a significant effect on the stomatal pore length, pore width, and pore area of melon seedlings, and the pore area showed a significant decreasing trend as the proportion of green light replacement increased (Table 2). It has been shown that exposing mature leaves to low light conditions triggers long-range signals that control stomatal development and leads to a decrease in stomatal density in young leaves [40]. Ma et al. showed that stomatal density would increase with the increase in the percentage of green light [30]. However, this phenomenon was not observed in this experiment, probably due to the fact that the plants were under drought stress, their growth and development were slow, and the treatment time was relatively short, so the phenomenon of stomatal density change was not observed.

### 3.4. Effect of Green Light Percentage on Transcriptome of Phytohormones in Melon Seedlings under Drought Stress

In response to drought-induced oxidative damage, plants have evolved many defense strategies. Phytohormones respond in a highly synergistic manner to signals stimulated by the external environment, which is essential for plant growth and development [41,42], and are involved in multiple signaling pathways, which help plants to maintain a certain level of resilience under suboptimal conditions [43,44]. In the untargeted metabolome assay for plants, the results showed large differences in ABA, JA, and SA in the two treatments, and the assay results for metabolites showed that the three phytohormones processed differential genes in three pathway nodes. In the ABA pathway, significant differential genes were seen in SnRK2 (MELO3C021918). Li et al. [45] showed that SnRK2.6 interacts with phyB and is involved in light-induced stomatal opening and has a negative role in red-light-induced stomatal opening, and it has been shown that the ABA receptor interacts with the SnRK2-type kinase and inhibits the PP2C phosphatase activity, releases active SnRK2 kinase phosphorylation, and activates SLAC1 channels, leading to reduced guard cell swelling and stomatal closure [46], and so SnRK2 plays a key role in the ABA-regulated stomatal opening and closing pathway. In this experiment, the intensity of red and blue light in the light plasma was subsequently reduced as the proportion of green light substitution increased. The transcription factor MYC2 is an important regulator of the JA signaling pathway and plays a crucial role in the regulation of plant growth and stress tolerance. Under light/dark conditions, stomatal closure in the lower epidermis of MYC2-silenced leaves was weaker, water loss was faster, and drought tolerance was lower compared with that of the control plant leaves [47], and the study of poplar by Xia et al. The authors of [48] showed that MYC2 regulates stomatal density and water use efficiency by targeting EPF2/EPFL4/EPFL9 and that overexpression of MYC2 reduces stomatal density and increases plant water use efficiency. It has been shown that the transcription factor MYC2 is involved in the development of Arabidopsis seedlings under blue light [49,50]. The presence of a significantly up-regulated differential gene (MELO3C018555) in MYC2 in this experiment indicated that stomatal opening was significantly affected by the JA pathway under different light-quality treatments. Previous studies have shown that MYC2 is involved in leaf stomatal development, and Ma et al. showed a significant change in stomatal density in a short-term drought treatment of cucumber seedlings in the treatment where green light replaced part of the red/blue light compared to the control treatment [30], but this phenomenon was not observed in this experiment. SA has been shown to ameliorate drought stress in plants [51,52] by inducing stomatal closure in intact leaves and directly controlling stomatal movement by increasing the levels of ROS and nitric oxide (NO) in the guard cells [53], and it has been shown that TAG, with light-induced stomatal opening, is mainly mediated by PHOT1 and PHOT2 blue light receptors [54], and the results of this experiment showed that green light instead of partial blue light treatment resulted in the presence of a significantly up-regulated differential gene (MELO3C011116) in TAG within the SA pathway, which enhanced the drought tolerance of melon seedlings. In addition to ABA, JA, and SA, T3 and CK treatments showed significant differential genes in multiple hormone signaling pathways of IAA, CK, GA, BR, and ETH (Figure 6c), which play important regulatory roles in plant drought stress [55,56,57,58,59], and all are regulated by light signaling [60,61,62,63,64]. Recently, it has been shown that green light is involved in the construction of plant photomorphology by regulating Arabidopsis hypocotyl elongation through BR [28]. In this study, the partial replacement of red and blue light treatments by green light resulted in significant differential genes at several nodes of the BR pathway in melon seedlings, including BKI1, BAK1, and BSK, and it is hypothesized that BR hormones play an important role in the green-light-induced regulation of stomata to elicit resistance responses.

### 3.5. Effect of Green Light Percentage on Phytohormone Metabolome of Melon Seedlings under Drought Stress

Drought stress treatments have significant effects on phytohormone metabolism [65,66], and changes in the light environment have a greater impact on plant metabolism, and we focused on differential metabolites in the pathway of phytohormone biosynthesis (ko01070). The results indicated that changes in the light environment mainly affected 13 differential metabolites. Among them, Aspartate, jasmonic acid, citric acid, Alpha-D-Glucose, tryptophan, and AMP differed significantly. When exposed to stress, plants accumulate large amounts of metabolites, especially amino acids. A series of studies have shown a close correlation between changes in Aspartate (Asp) content and plant stress [67,68,69]. And, the degree of opening of plant stomata affects the accumulation of ASP, as changes in its level may also be related to malic acid activity [70,71]. Ma et al. showed that increasing the proportion of green light in the plant’s light environment under drought stress affected the stomatal opening of cucumber seedlings and down-regulated the transcript levels of Aluminum-Activated Malate Transporter 9 (CsALMT9) [30], so the results of our experiments indicated that the addition of the green light treatment for JA is involved in a variety of abiotic stresses, including drought stress [72,73]. JA pretreatment significantly increased drought tolerance in barley [74], and JA increased drought tolerance in cauliflower by activating the enzymatic antioxidant system [75]. Our experimental results showed that JA content was significantly increased by adding green light treatment. Citric acid plays a key role in the initial stage of the TCA cycle. When present in excess, citric acid inhibits glycolysis, stimulates gluconeogenesis, and hinders downstream TCA reactions. Plants assume “dormancy” to prevent cells from wasting large amounts of energy and to ensure faster recovery under normal conditions [76]. FRANCO et al. showed that citric acid accumulation correlates with plant photosynthesis and that at high PPFD, leaves of Clusia accumulated large amounts of malic and citric acids [77]. Plants did not absorb green light, and the intensity of light (red and blue) effective for plant photosynthesis decreased under the T3 treatment compared to the control. Therefore, we found that the citric acid content was reduced under the T3 treatment compared to the CK group. Alpha D-glucose is a common reducing monosaccharide. The results of Lu et al. showed that the content of reducing sugars increased in rice under drought conditions and the content of reducing monosaccharides such as erythrulose, mannose, alpha D-glucose, and galactose increased in the metabolism group, and a variety of monosaccharides participated in the redox reactions of plants as cofactors and antioxidants. The results of this experiment showed a decrease in alpha D-glucose content under the T3 treatment compared to the CK treatment, indicating that the addition of green light reduced the degree of drought stress in plants [78]. When plants are subjected to stress, they will accumulate more tryptophan, and as tryptophan is a target of oxidation, the free amino acid may provide a buffer between ROS and proteins in the chloroplast where both tryptophan and ROS are synthesized [79]. Our results showed that the decrease in tryptophan content under the T3 treatment compared to the CK treatment could be attributed to the addition of green light reducing the stress level of seedlings, resulting in a decrease in the content of O_2_^∙−^ and H_2_O_2_, the main members participating in redox reactions. PAP has been reported to be a secondary messenger of ABA signaling that regulates the response to drought and oxidative stress [80], and AMP is a by-product of PAP that plays a role as a monomer in RNA production. It is a component of many metabolic processes [81]. The results of this experiment showed a significant increase in AMP content under the CK treatment compared to the T3 treatment, indicating a more verified drought stress.

## 4. Materials and Methods

### 4.1. Experimental Materials

This experiment was conducted in 2022 in the Biological and Environmental Engineering Laboratory of Facility Agriculture, College of Horticulture, Northwest A&F University. The test material was melon ‘Emerald’ (*Cucumis melo*), and the seeds were purchased from Shandong Zibo Harvest Seed Industry Science and Technology Co. Melon seeds were soaked in distilled water for 8 h and then germinated in the dark for 48 h. After germination, they were sown in 10 cm × 10 cm nutrient pots and grown in an artificial climatic chamber with a light intensity of 150 μmol/(m^2^·s), day/night temperatures of 25 °C/20 °C, a photoperiod of 12 h/12 h, and a relative humidity of 60%, and were watered with 1/4 Hoagland melon nutrient solution (pH 6.5 ± 0.1, EC of 2.2–2.5 ms/cm) until the substrate reached water holding saturation.

### 4.2. Experimental Design

After growing to 3 leaves and 1 heart for about 20 d in the artificial climate chamber, the seedlings with consistent growth were selected and transplanted into a cultivation frame equipped with an LED light source, in which the day and night temperatures were 28 °C/25 °C, relative humidity was 40%~50%, and drought stress was applied in the form of no watering, with a total of 9 d for the drought treatment.

We controlled the light quality through the LED light board; the LED light board consists of different light qualities of LED lamp beads, with a light board connected to the controller that can independently control the intensity of each light quality and light time. LED light panels provide a specific spectrum of light, and the quality, duration and intensity of the LED light is controlled through a controller.Optical panels and controllers manufactured by the Yinbian company (Xi’an, Shaanxi, China) Reflective films were posted all around the LED panels to ensure uniform radiation in all places. The PPFD of all treatments was controlled to be 250 μmol/(m^2^·s), the red-to-blue ratio (R/B) was guaranteed to be 4:1 for all treatments, and the proportion of green light (G) was adjusted (Table 3). The spectrogram under each treatment is shown in Figure 7.

### 4.3. Measuring Items and Methods

#### 4.3.1. Morphometric Indicators

After 9day of treatment, five plants were randomly selected and their aboveground height was determined using a scale; stem thickness was determined using vernier calipers; the area of each leaf was determined using a leaf area meter (AM350 Portable Leaf Area Meter, Hoddesdon, UK); the aboveground fresh weight was determined using an electronic balance; and the aboveground weight of the plants was determined by placing the plants in the oven at 105 °C for 15 min after being stored in the oven and then dried at 80 °C to a constant weight for the determination of their dry weights.

#### 4.3.2. Measurement of Photosynthetic Properties

Measurements of gas exchange were carried out on day 3, day 6, and day 9 of the treatments. Five seedlings were randomly selected from each treatment, and their 2nd true leaves were taken to determine the transpiration rate (E), net photosynthetic rate (Pn), intercellular CO_2_ concentration (Ci), and stomatal conductance (gsw) using Plant Photosynthesizer 6800 (LI-6800, Lincoln, NE, USA). The temperature of the leaf chamber was set at 24 °C, the CO_2_ level at 400 µmol/mol, and the relative humidity at 60%, and the light source for the determination was set to be R90B10 with a light intensity of 1000 µmol/(m^2^·s).

#### 4.3.3. Stomatal Morphology and Electrolyte Permeability (REC)

The stomatal pore size was observed on the 9th day of treatment. Three seedlings were randomly selected from each treatment, and the same leaf position of the 2nd leaf was taken to make clinical mounts; the taken leaves were placed on transparent adhesive tape, lightly pressed to make a tight bond, and then lightly scraped with a razor blade to remove the chloroplastic tissues, and the mounts were pasted to the slides. The sections were observed under a microscope (Olympus BX63, Olympus LS, Tokyo, Japan), and stomatal size was measured at 40×. Ten fields of view were selected for each mount, and 20 stomata were selected for each field of view. Pore length and pore width were measured using the ImageJ software (64-bit), and pore area was calculated. Pore area (μm^2^) = π × pore length/2 × pore width/2.

The 2nd leaf was selected for each treatment and leaf discs of 1 cm diameter were made using a hole punch in the remaining leaf parts avoiding the main leaf veins. Ten leaf discs were placed in a test tube containing 10 mL of distilled water and immersed in darkness at room temperature for 4 h, during which they were shaken 3–5 times, after which their conductivity was determined as S1 by a conductivity meter, and then the tubes were water-bathed at 100 °C for 20 min, and then shaken after cooling, and then the conductivity was determined as S2. Relative conductivity (%) = (S1 − S0)/(S2 − S0) × 100%, where S0 is the distilled water’s conductivity. 

#### 4.3.4. Measurement of Malondialdehyde (MDA) and Superoxide Dismutase (SOD) Enzyme Activity

The MDA content was determined by a spectrophotometer using an MDA content assay kit (Solebo, Beijing, China). Weigh 0.1 g of tissue and add 1 mL of extraction solution for ice bath homogenization, centrifuge at 8000× *g* for 10 min at 4 °C, take the supernatant, and add the reagents sequentially according to the procedure of the manual in a 100 °C water bath for 60 min and then 10,000× *g*. After centrifugation at room temperature for 10 min, measure the absorbance at 532 nm and 600 nm, and calculate the MDA content according to the manual. 

The content of the SOD activity was determined by a spectrophotometer using a SOD activity assay kit (Solebo, Beijing, China). Weigh 0.1 g of tissue and add 1 mL of extraction solution for ice bath homogenization, centrifuge at 8000× *g* for 10 min at 4 °C, take the supernatant, add the reagents according to the steps in the instructions in sequence, measure the absorbance at 560 nm after 30 min in a 37 °C water bath, and calculate the SOD activity according to the instructions.

#### 4.3.5. Determination of Hydrogen Peroxide (H_2_O_2_) and Superoxide Anion (O_2_^∙−^) Content

The H_2_O_2_ content was determined by a spectrophotometer using an H_2_O_2_ content assay kit (Solebo, Beijing, China). Weigh 0.1 g of tissue and add 1 mL of extraction solution for ice bath homogenization, centrifuge at 8000× *g* at 4 °C for 10 min, take the supernatant, add the reagents sequentially according to the steps in the instruction manual, measure the absorbance at 415 nm, and calculate the H_2_O_2_ content according to the instruction manual.

The O_2_^∙−^ content was determined by a spectrophotometer using an O_2_^∙−^ content assay kit (Solebo, Beijing, China). Weigh 0.1 g of tissue and add 1 mL of extraction solution for ice bath homogenization, centrifuge at 12,000 rpm and 4 °C for 20 min, take the supernatant, add the reagents sequentially according to the instruction, measure the absorbance at 530 nm, and calculate the O_2_^∙−^ content according to the instruction.

On the 9th day of the treatments carried out, the 2nd true leaves of three melon seedlings from each treatment were selected and sampled for tissue staining. DAB staining and NBT staining were referred to by Hu et al. [82].

#### 4.3.6. RNA Extraction and Transcriptome Sequencing

The 2nd leaf of the melon seedlings under CK and T3 treatments was taken after 9 d of drought treatment, and the total RNA was extracted using Trizol reagent (Thermofisher, Waltham, MA, USA) according to the steps provided by the manufacturer. After extraction of the total RNA, mRNA was purified using Dynabeads Oligo (dT) (Thermofisher, Waltham, MA, USA) from the total RNA (5 µg). The total RNA extracted from each leaf sample was used for RNA-Seq library construction and sequencing by Lianchuan Biotechnology Co. (Beijing, China). After purification, mRNA was fragmented into short fragments using divalent cations at a high temperature using RNA Fragmentation Module (NEB, cat. e6150, Ipswich, MA, USA) at 94 °C for 5–7 min. Then, cDNA was generated by reverse transcription with SuperScript™ II reverse transcriptase (Invitrogen, cat. 1896649, Waltham, MA, USA) (NEB, cat. m0209, Ipswich, MA, USA), RNase H (NEB, cat. m0297, Ipswich, MA, USA), and dUTP solution (Thermofisher, cat. R0133, Waltham, MA, USA). U-labeled second-strand DNA was synthesized with R0133 (Thermofisher, cat. R0133, Waltham, MA, USA) The ligated products were amplified by PCR after the treatment of U-labeled second-strand DNA with the thermal UDG enzyme (NEB, cat. m0280, USA). The average insert fragment size of the final cDNA library was 300 ± 50 bp for the average insert length of the cDNA library. Finally, 2 × 150 bp paired-end sequencing (PE150) was performed on IlluminaNovaseq™6000 (LC-Bio Technology CO., Hangzhou, China).

#### 4.3.7. Metabolomics Analysis and Differential Metabolite Identification

The 2nd leaves of melon seedlings under the CK and T3 treatments were taken after 9 days of drought treatment. A total of 100 μL of the sample was taken and mixed with 400 μL of extraction solution (MeOH:ACN, 1:1(*v*/*v*)); the extraction solution contained deuterated internal standards, and the mixed solution was vortexed for 30 s, sonicated for 10 min in 4 °C water bath, and incubated for 1 h at 40 °C to precipitate proteins. Then, the samples were centrifuged at 12,000 rpm (RCF = 13,800× *g*, R = 8.6 cm) for 15 min at 4 °C. The supernatant was transferred to a fresh glass vial for analysis. The quality control (QC) sample was prepared by mixing an equal aliquot of the supernatant of the samples. All samples were collected by an LC-MS system following machine commands. All chromatographic separations were performed using an UltiMate 3000UPLC system (Thermofisher, Waltham, MA, USA) with a reversed-phase separation on an ACQUITY UPLCT3 column (100 mm × 2.1 mm, 1.8 µm, Waters, Milford, MA, USA). The metabolites eluting from the column were detected using a high-resolution tandem mass spectrometer TripleTOF 6600 (SCIEX, Carlsbad, CA, USA). The collected mass spectrometry data were preprocessed using XCMS software (4.7). It was then processed with the XCMS, CAMERA, and metaX toolboxes implemented in the R software (4.2.1). The ions were identified by combining retention time (RT) and *m*/*z* data. The metabolites were annotated using the online KEGG and HMDB databases, and the exact molecular mass data (*m*/*z*) of the samples were matched to the data in the database. Statistical analysis of the data was mainly conducted using the R software (version4.0); a clustering heatmap was plotted by the R package pheatmap; PCA analysis and significantly different protein analyses were conducted using the R package metaX (2.67); PLSDA analysis was performed using the R package ropls, and VIP values of each variable were calculated; correlation analysis was performed using the Pearson correlation coefficient of R package cor, and a *p*-value < 0.001 was calculated using a T-test obtained by a *p*-value < 0.05 and a multiplicity with a difference > 1.2; and the VIP calculated using PLSDA analysis was satisfied simultaneously to screen the final significantly different metabolites. Differential enrichment analysis of the KEGG pathway was performed based on the hypergeometric test, and the functional entries with a *p*-value < 0.05 from the statistical test were the functional entries significantly enriched for differential proteins.

#### 4.3.8. Real-Time Quantitative RT-PCR Assay

In total, 2 g of leaf tissue was taken into a 2 mL centrifuge tube and quickly frozen in liquid nitrogen, followed by high-frequency shaking using a high-throughput grinder. The reliability of RNA-Seq data was verified by qRT-PCR using nine randomly selected differential genes. Primers were designed by Primer Premier 5.0, and the primer sequences are shown in Appendix A. The total leaf RNA was extracted using the RBA extraction kit (Aidab, Beijing, China) according to the standard step-by-step method of the reagent. A reverse transcription kit (Cofitt, Hangzhou, China) was used. The extracted RNA was reverse transcribed into cDNA using a reverse transcription kit (Cofitt, Hangzhou, China) according to the standard method of the reagent, and the relative expression of the genes was analyzed using the 2^−ΔΔCt^ analysis method in a 20 μL reaction system according to the 2× qPCR SmArt Mix (SYBR Green) kit (Descartes Biologicals, Shanghai, China).

### 4.4. Statistical Analysis

Data were collated and plotted using Microsoft Office Excel 2020 and analyzed for significant differences using IBM SPSS Statistics 25.

## 5. Conclusions

On the basis of determining the ratio of red and blue light and photosynthetically active radiation, increasing the ratio of green light instead of red and blue light improved the drought tolerance of melon seedlings; at the same time, it reduced the electrolyte permeability and MDA content of melon seedling leaves and increased O_2_^∙−^ and H_2_O_2_ content, which helped to alleviate the damage of short-term drought to the cell membranes of melon seedlings. The addition of green light initially decreased leaf stomatal conductance and reduced water loss. The transcriptomic and metabolomic results indicated that the process of green light-regulated enhancement of drought tolerance in plants is regulated by multiple phytohormones. This study provides a new perspective for thinking about improving plant water utilization in facilities.

## Figures and Tables

**Figure 1 ijms-25-07561-f001:**
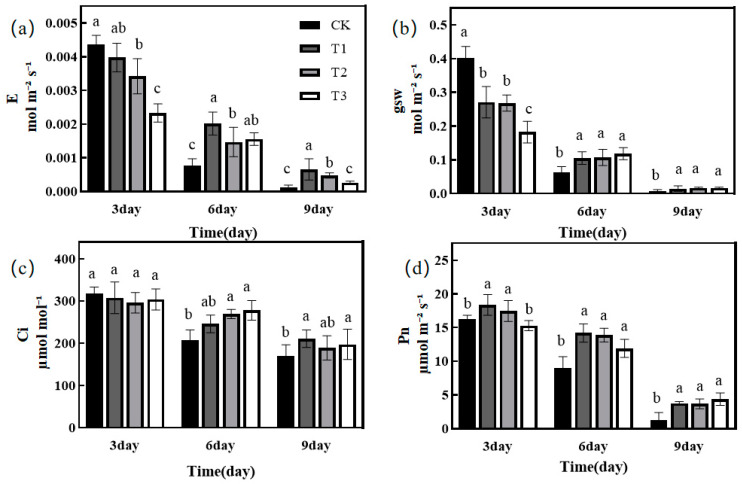
Effect of different treatments on photosynthetic characteristics of melon seedlings. (**a**) Effect of transpiration rate (E) between treatments; (**b**) effect of stomatal conductance (gsw) between treatments; (**c**) effect of intercellular CO_2_ (Ci) between treatments; and (**d**) effect of net photosynthetic rate (Pn) between treatments. Note: in same figure, different letters represent significant differences (*p* < 0.05), and same letters represent insignificant differences (*p* > 0.05).

**Figure 2 ijms-25-07561-f002:**
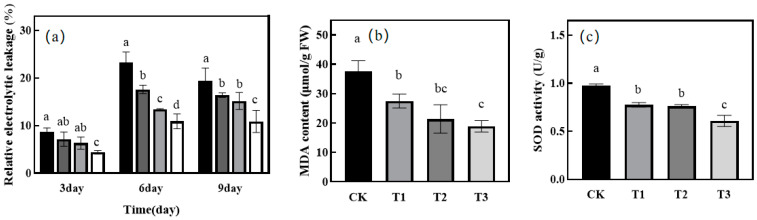
Effect of different treatments on relative electrolytes, MDA content, and SOD activity in melon seedlings. (**a**) Relative electrolytic leakage under different treatments measured on 3, 6, and 9 days. (**b**) MDA content under different treatments at 9 days. (**c**) SOD content under different treatments at 9 days. Different letters represent significant differences (*p* < 0.05), while the same letters represent no significant differences (*p* > 0.05).

**Figure 3 ijms-25-07561-f003:**
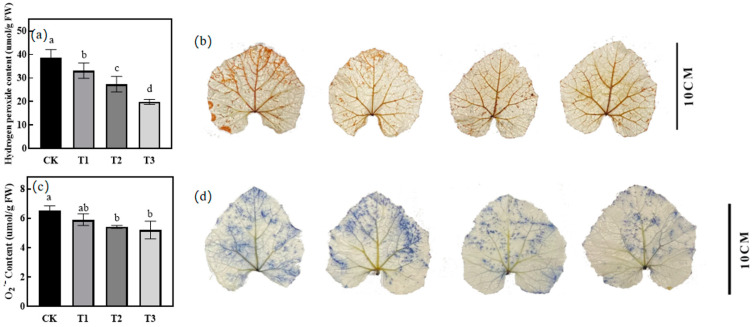
Effect of different treatments on ROS in melon seedlings. (**a**) H_2_O_2_ content under different treatments at 9 days. (**b**) DAB staining of melon seedling leaves under each treatment at 9 days. (**c**) O_2_^∙−^ content under different treatments at 9 days. (**d**) NBT staining of melon seedling leaves under each treatment at 9 days. Different letters represent significant differences (*p* < 0.05), while the same letters represent no significant differences (*p* > 0.05).

**Figure 4 ijms-25-07561-f004:**
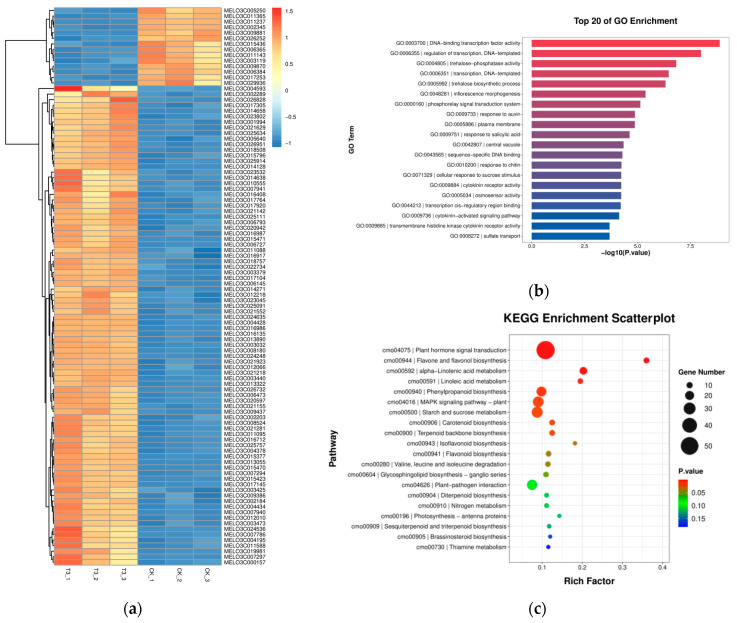
RNA-Seq-based analysis of DEGs in melon seedlings under different treatments. (**a**) RNA-Seq-based heat map analysis of DEGs under different treatments in melon seedlings. (**b**) GO functional classification of DEGs in different treatments under drought stress. (**c**) Classification of KEGG pathways of DEGs in different treatments under drought stress.

**Figure 5 ijms-25-07561-f005:**
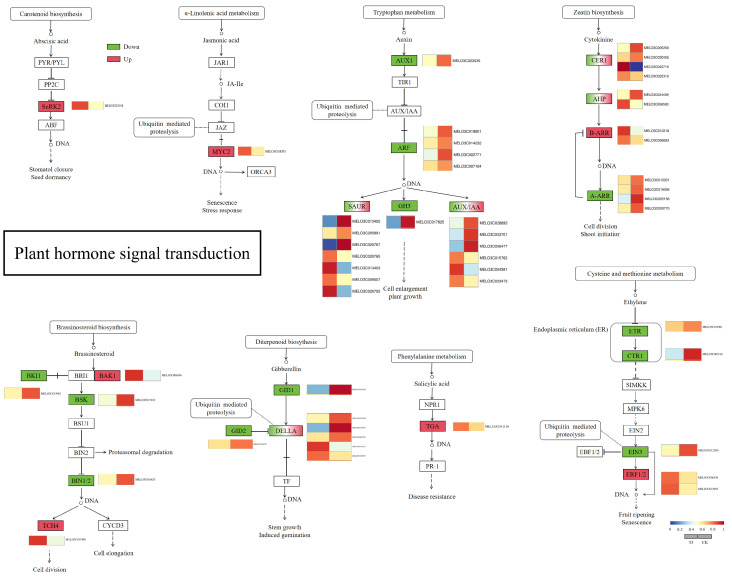
Analysis of plant hormone signal transduction pathways.

**Figure 6 ijms-25-07561-f006:**
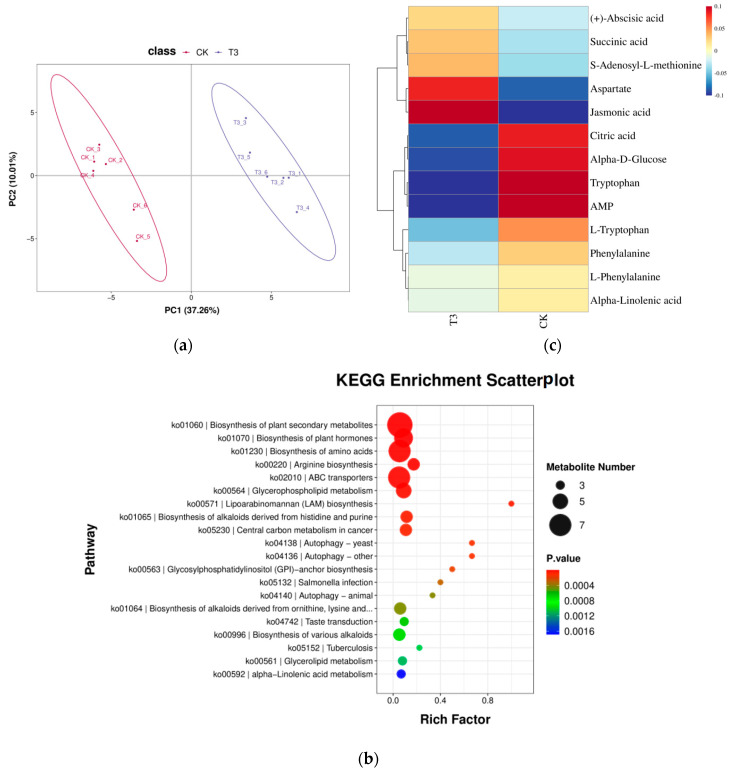
LC-MS-based analysis of differential metabolites in melon seedlings under different treatments. (**a**) Metabolite pattern diagram for PCA analysis. (**b**) Classification of KEGG pathways of different treatment differential metabolites under drought. (**c**) Changes in metabolites related to phytohormone synthesis after different treatments.

**Figure 7 ijms-25-07561-f007:**
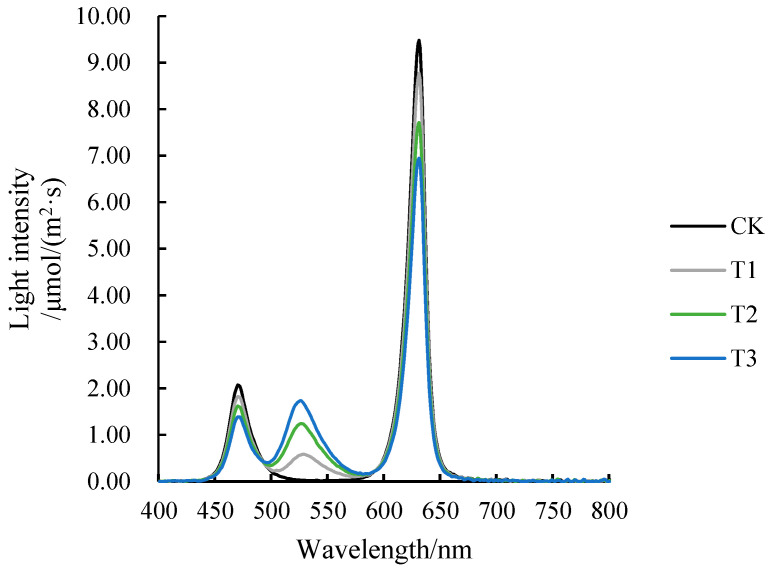
Spectrograms of different treatments.

**Table 1 ijms-25-07561-t001:** Effects of different treatments on morphological indices of melon seedlings.

Treatment	Plant Height	Stem Diameter	Fresh Weight	Dry Weight	Total Leaf Area
CK	13.633 ± 1.075 c	4.060 ± 0.141 a	6.503 ± 0.341 c	0.723 ± 0.003 b	138.563 ± 7.467 c
T1	13.827 ± 0.693 c	4.237 ± 0.289 a	6.259 ± 0.655 c	0.853 ± 0.100 ab	168.807 ± 6.289 b
T2	15.487 ± 0.365 b	4.137 ± 0.065 a	7.564 ± 0.138 b	0.899 ± 0.090 ab	170.253 ± 5.830 b
T3	17.573 ± 0.552 a	4.420 ± 0.358 a	10.112 ± 0.210 a	0.941 ± 0.1306 a	231.680 ± 15.790 a

Note: Within the same column, different letters represent significant differences (*p* < 0.05), while the same letters represent no significant differences (*p* > 0.05). The same is below.

**Table 2 ijms-25-07561-t002:** Effects of different treatments on stomatal characteristics of melon seedling leaves.

Treatment	Pore Width/μm	Pore Length/μm	Stomatal Width/μm	Stomatal Length/μm	Pore Area/μm^2^
CK	13.633 ± 1.075 c	4.060 ± 0.141 a	6.503 ± 0.341 c	0.723 ± 0.003 b	138.563 ± 7.467 c
T1	13.827 ± 0.693 c	4.237 ± 0.289 a	6.259 ± 0.655 c	0.853 ± 0.100 ab	168.807 ± 6.289 b
T2	15.487 ± 0.365 b	4.137 ± 0.065 a	7.564 ± 0.138 b	0.899 ± 0.090 ab	170.253 ± 5.830 b
T3	17.573 ± 0.552 a	4.420 ± 0.358 a	10.112 ± 0.210 a	0.941 ± 0.1306 a	231.680 ± 15.790 a

**Table 3 ijms-25-07561-t003:** Intensity of red light, blue light, and green light under different treatments.

Treatment	Red Light μmol/(m^2^·s)	Blue Light μmol/(m^2^·s)	Green Light μmol/(m^2^·s)	PPFD μmol/(m^2^·s)	R:B:G
CK	200	50	0	250	4:1:0
T1	182	45	23	250	4:1:0.5
T2	160	40	50	250	4:1:1.25
T3	143	36	71	250	4:1:2

## Data Availability

Data is contained within the article.

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
