# Peer review of "Effect of Green Light Replacing Some Red and Blue Light on Cucumis melo under Drought Stress"

_ijms, 2024, doi:10.3390/ijms25147561_

Round 1
Reviewer 1 Report
Comments and Suggestions for Authors
The submitted manuscript addresses the issue of the effect of light on physiological parameters of juvenile melon plants. This is a topical issue as the effect of light is also addressed with respect to energy savings and production quality obtained from controlled conditions. The manuscript is written with relative care. However, minor typos in the text need to be corrected. I recommend that the abstract be edited as it is too general. I recommend not only to give the measured values but also to specify the individual variants. The designation T3 is not explained when it is first mentioned, but only later. It is not clear in the methodology how the plants were grown at the 3 leaf stage. Please add if I have not overlooked. The results are described in relative values and not in measured values. Please correct. The quality of graphs 1-6 is not the best. The graphs are small, hard to read. The discussion is rather descriptive. I recommend its revision.
Author Response
Response to Reviewer 1 Comments
Thank you very much for taking the time to review this manuscript. Please find the detailed responses below and the corresponding revisions/corrections highlighted/in track changes in the re-submitted files.
A brief summary:
The submitted manuscript addresses the issue of the effect of light on physiological parameters of juvenile melon plants. This is a topical issue as the effect of light is also addressed with respect to energy savings and production quality obtained from controlled conditions. The manuscript is written with relative care. However, minor typos in the text need to be corrected.
Q1:I recommend that the abstract be edited as it is too general. I recommend not only to give the measured values but also to specify the individual variants.
A1:We have re-edited the abstract.
Q2:The designation T3 is not explained when it is first mentioned, but only later.
A1:We have re-edited the abstract and corrected the problem
Q3:It is not clear in the methodology how the plants were grown at the 3 leaf stage. Please add if I have not overlooked.
A3:We describe the growing conditions in the manuscripts”the seedlings with consistent growth were selected and transplanted into a cultivation frame equipped with LED light source, in which the day and night temperatures were 28℃/25℃, relative humidity was 40%~50%, and drought stress was applied in the form of no watering, with a total of 9 d drought treatment”.
Q4:The results are described in relative values and not in measured values. Please correct.
A4:We have used relative values to describe the results.
Q5:The quality of graphs 1-6 is not the best. The graphs are small, hard to read.
A5:We replaced these graphs.
Q6:The discussion is rather descriptive. I recommend its revision.
A6:We have rewritten part of the discussion and added some missing content (metabolomics).
Reviewer 2 Report
Comments and Suggestions for Authors
Essay
Effect of Green Light Replacing Some Red and Blue Light on Cucumis melo L. under Drought Stress
A brief summary
The aim of the work seems interesting but the methodology used is questionable. The text also requires various editorial corrections.
Broad comments
1. The study examined how light quality - the introduction of green light in addition to red and blue light - affects the response of melon plants to drought stress, including morphometric, photosynthetic properties, stomatal morphology, enzyme activity, hydrogen peroxide and superoxide anion content, transcriptome analysis including plant hormone synthesis pathways and metabolomic response.
2. The multifaceted approach to the subject is valuable and distinctive. This allows for a holistic view of the issue under study. Unfortunately, this has not been fully exploited.
3. Most interesting are the sections on transcriptome analysis, including plant hormone synthesis pathways, and metabolomics, but these are not fully discussed and explained. Especially metabolomics, which is actually not discussed and summarised at all and does not even have its own subsection in the Discussion.
4. About lights. The title is not precise. The title refers to ‘green light REPLACING SOME red and blue light’, while Table 1 and Figure 1 show that blue and red light was still present and that the proportion of green light in PAR (200 um/m2 s) is increasing.
a. The red-to-blue-to-green ratio shown in the last column of Table 1 (R:B:G) is also inaccurate because it does not add up. What is the ratio to? The number of LEDs per unit area? What was the source of the green light? What diodes were used? What blue and red LEDs were used? Were four different LED panels therefore used?
b. Why is a PPFD value (250 um/m2 s) shown in the text and a PAR value (200 um/m2 s) shown in the table?
c. So in summary: REPLACING SOME particular diodes, or REPLACING SOME light energy? But how about ENRICHMENT or ‘increasing the proportion of green light’ like in Conclusion?
5. The Conclusions summarise in an appropriate way the results obtained but are presented very compactly. They are actually 3 sentences, with the last sentence not directly addressing the results.
6. The references appear to be appropriately selected, but need to be corrected according to the journal's requirements.
7. Additional comments and suggestions can be found below.
Specific comments
1. About the Abstract. The order in which abbreviations and explanations appear is incorrect.
a. First 'CK' and 'T3' appear in line 19 and only in line 26 'control group (CK)' and 'addition of green light treatment (T3)'.
b. Similarly, 'Cucumis melo' appears in line 14 (should be in italics) and 'menon' appears in line 21.
2. Keywords unnecessarily repeat those already used in the title. It is better to use 'melon',' drought resistance', 'light quality'.
3. The notation 'd' should probably be replaced by 'day/s', however. This is not a generally recognized abbreviation/notation.
4. Lots of typos, especially missed spaces after full stops, failure to use capital letters at the beginning of sentences, forms such as ‘PValue’, ‘Fig.2’ ect.
5. Line 201. Please explain/describe ‘… metabolites were extracted with 50% methanol buffer.’.
6. Table 2. Please add the type of post-hoc test, the level of significance (α), the number of replicates (n) and what the small letters next to the bars stand for - what is being compared to what.
7. Figure 2. Please add the type of post-hoc test, the level of significance (α), the number of replicates (n) and what the small letters next to the bars stand for - what is being compared to what. Titles of individual graphs - the measured parameters - can also be added.
8. Table 3. Please add the type of post-hoc test, the level of significance (α), the number of replicates (n) and what the small letters next to the bars stand for - what is being compared to what.
9. Figure 3 and 4. Please add the type of post-hoc test, the level of significance (α), the number of replicates (n) and what the small letters next to the bars stand for - what is being compared to what. Titles of individual graphs - the measured parameters - can also be added. Which day of observance is it?
10. Subheadings 3.7.1, 3.7.2 and 3.7.3 are identical and do not relate to the content.
11. Figures 5, 6 and are very poorly legible - a resolution is too low.
12. Lines 587-488 - repeated sentences?
13. Lines 510-517 The sentence is horrendously long and the end is repeated twice.
Author Response
Response to Reviewer 2 Comments
Thank you very much for taking the time to review this manuscript. Please find the detailed responses below and the corresponding revisions/corrections highlighted/in track changes in the re-submitted files.
Broad comments
Q1:(a) The study examined how light quality - the introduction of green light in addition to red and blue light - affects the response of melon plants to drought stress, including morphometric, photosynthetic properties, stomatal morphology, enzyme activity, hydrogen peroxide and superoxide anion content, transcriptome analysis including plant hormone synthesis pathways and metabolomic response.
(b) The multifaceted approach to the subject is valuable and distinctive. This allows for a holistic view of the issue under study. Unfortunately, this has not been fully exploited.
(c)Most interesting are the sections on transcriptome analysis, including plant hormone synthesis pathways, and metabolomics, but these are not fully discussed and explained. Especially metabolomics, which is actually not discussed and summarised at all and does not even have its own subsection in the Discussion.
A1:we have made some additions in the latest manuscript.
Q2:About lights. The title is not precise. The title refers to‘green light REPLACING SOME red and blue light’, while Table 1 and Figure 1 show that blue and red light was still present and that the proportion of green light in PAR (200 um/m2⋅s) is increasing.
A2:Red and blue light is necessary for plant growth, our experiment controls the ratio of red and blue light to be 4:1 and increases the proportion of green light, hence the title "Green light replaces some red and blue light".
Q3. The red-to-blue-to-green ratio shown in the last column of Table 1 (R:B:G) is also inaccurate because it does not add up. What is the ratio to? The number of LEDs per unit area? What was the source of the green light? What diodes were used? What blue and red LEDs were used? Were four different LED panels therefore used?
A3:The ratio of R:B:G is determined by the intensity of each light quality, and the reason we list the ratio is to show that although the percentage of green light is increasing, the ratio of R:B has been fixed at 4:1. We used 4 sets of light qualities, whose spectrograms are shown in FIG. 1.
Q4. Why is a PPFD value (250 um/m2 s) shown in the text and a PAR value (200 um/m2 s) shown in the table?
A2:This is a revision error, the correct PAR is 250, which we have corrected in the latest manuscript.In this experiment the PPDF values were the same as the PAR values and had a similar significance
Q5. So in summary: REPLACING SOME particular diodes, or REPLACING SOME light energy? But how about ENRICHMENT or‘increasing the proportion of green light’like in Conclusion?
A5:The manuscript meant to‘increasing the proportion of green light’, and we have corrected this misrepresentation.
Q6. The Conclusions summarise in an appropriate way the results obtained but are presented very compactly. They are actually 3 sentences, with the last sentence not directly addressing the results.
A6:We re-edited the conclusions.
Q7. The references appear to be appropriately selected, but need to be corrected according to the journal's requirements.
A7:We re-edited the reference format.
Specific comments
Q1. About the Abstract. The order in which abbreviations and explanations appear is incorrect.
- First 'CK' and 'T3' appear in line 19 and only in line 26 'control group (CK)' and 'addition of green light treatment (T3)'.
- Similarly, 'Cucumis melo' appears in line 14 (should be in italics) and 'menon' appears in line 21.
A1:We re-ediedt the Abstract and solved the problem.
Q2. Keywords unnecessarily repeat those already used in the title. It is better to use 'melon',' drought resistance', 'light quality'.
A2:We've taken your advice and modified keywords.
Q3. The notation 'd' should probably be replaced by 'day/s', however. This is not a generally recognized abbreviation/notation.
A7:We re-edited the notation.
Q4. Lots of typos, especially missed spaces after full stops, failure to use capital letters at the beginning of sentences, forms such as ‘PValue’, ‘Fig.2’ ect.
A4:We solved the problem.
Q5. Line 201. Please explain/describe ‘… metabolites were extracted with 50% methanol buffer.’
A5:There are some omissions about this method, which we have added in the latest manuscript.
“100 μL of sample was taken, mixed with 400 μL of extraction solution (MeOH:ACN, 1:1
(v/v)), the extraction solution contain deuterated internal standards, the mixed solution
were vortexed for 30 s, sonicated for 10 min in 4 ℃ water bath, and incubatedfor 1 h at
-40 ℃ to precipitate proteins. Then the samples were centrifuged at 12000 rpm
(RCF=13800(×g),R= 8.6cm) for 15 min at 4 ℃. The supernatant was transferred to afresh
glass vial for analysis. The quality control (QC) sample was prepared by mixing an equal aliquot
of the supernatant of samples.”
Q6. Table 2. Please add the type of post-hoc test, the level of significance (α), the number of replicates (n) and what the small letters next to the bars stand for - what is being compared to what.
A6:We added the instruction.
Q7. Figure 2. Please add the type of post-hoc test, the level of significance (α), the number of replicates (n) and what the small letters next to the bars stand for - what is being compared to what. Titles of individual graphs - the measured parameters - can also be added.
A7:We added the instruction in the latest manuscript.
Q8. Table 3. Please add the type of post-hoc test, the level of significance (α), the number of replicates (n) and what the small letters next to the bars stand for - what is being compared to what.
A6:We added the instruction in Table 2,and added the sentence “ The same below”,it also suit Table3.
Q9. Figure 3 and 4. Please add the type of post-hoc test, the level of significance (α), the number of replicates (n) and what the small letters next to the bars stand for - what is being compared to what. Titles of individual graphs - the measured parameters - can also be added. Which day of observance is it?
A9:We added the instruction in the latest manuscript.
Q10. Subheadings 3.7.1, 3.7.2 and 3.7.3 are identical and do not relate to the content.
A10:We re-edited the subheadings.
Q11. Figures 5, 6 and are very poorly legible - a resolution is too low.
A11:We replaced it with a higher quality image.
Q12. Lines 587-488 - repeated sentences?
A12:We re-edited the sentences.
Q13. Lines 510-517 The sentence is horrendously long and the end is repeated twice.
A13:We re-edited the conclusion and solve the problem.
Round 2
Reviewer 2 Report
Comments and Suggestions for Authors
The authors have done a considerable amount of work and have satisfactorily addressed most of the comments suggested.
However, not all of them:
- It is still unclear what diodes were used and how the lamps were designed.
- Figures 5 and 6 are still on the edge of legibility.
- Missing spaces and other typos will certainly be corrected in the later stages of preparing the mauscript.
Author Response
Response to Reviewer Comments
Thank you very much for taking the time to review this manuscript. Please find the detailed responses below and the corresponding revisions/corrections highlighted/in track changes in the re-submitted files.
Q1:It is still unclear what diodes were used and how the lamps were designed.
A1:Through the LED light board to control the quality of light, LED light board as shown in the figure, by four kinds of different light quality (red, blue, green and far red) LED lamp beads, the light board connected to the controller can be independently controlled by the intensity of each kind of light quality and light time.
Q2:Figures 5 and 6 are still on the edge of legibility.
A2:We replaced it with a higher quality image.
Q3:Missing spaces and other typos will certainly be corrected in the later stages of preparing the mauscript.
A3:We rechecked and corrected the spelling errors and missing spaces.
